# High accuracy gene expression profiling of sorted cell subpopulations from breast cancer PDX model tissue

**Warren Porter[☉], Eileen Snowden[☉], Friedrich Hahn, Mitchell Ferguson, Frances Tong, W. Shannon Dillmore, Rainer Blaesius[ID]***

BD Technologies and Innovation, Research Triangle Park, NC, United States of America

[☉] These authors contributed equally to this work.

* Rainer.Blaesius@bd.com

**Data Availability Statement:** All data are within the manuscript and supporting information files. The complete RNASeq raw data can be accessed at the EMBL-EBI ArrayExpress server under the

## Abstract

Intratumor Heterogeneity (ITH) is a functionally important property of tumor tissue and may be involved in drug resistance mechanisms. Although descriptions of ITH can be traced back to very early reports about cancer tissue, mechanistic investigations are still limited by the precision of analysis methods and access to relevant tissue sources. PDX models have provided a reproducible source of tissue with at least a partial representation of naturally occurring ITH. We investigated the properties of phenotypically distinct cell populations by Fluorescence activated cell sorting (FACS) tissue derived cells from multiple tumors from a triple negative breast cancer patient derived xenograft (PDX) model. We subsequently subjected each population to in depth gene expression analysis. Our findings suggest that process related gene expression changes (caused by tissue dissociation and FACS sorting) are restricted to Immediate Early Genes (IEGs). This allowed us to discover highly reproducible gene expression profiles of distinct cellular compartments identifiable by cell surface markers in this particular tumor model. Within the context of data from a previously published model our work suggests that gene expression profiles associated with hypoxia, stemness and drug resistance may reside in tumor subpopulations predictably growing in PDX models. This approach provides a novel opportunity for prospective mechanistic studies of ITH.

## Introduction

Phenotypic differences between patients with the same tumor type (intertumor heterogeneity) and among cancer cells within the same patient (intratumor heterogeneity/ITH) have always been integral to descriptions of tumor appearance [1], yet they were neglected as a source of useful information for a long time. While intertumor heterogeneity has been integrated into clinical practice, particularly in breast cancer [2], ITH remains to be included as a standard part of cancer tissue analysis. The technical difficulty to assess ITH has hindered efficient ways to build on observations of inherent complexity described in the 1970s and 80s and arguably a general resistance to this concept [3] prevented an unbiased appreciation. The past 10–15 years have seen a "re-discovery" of ITH as a mainstream topic. Models of "tumor as an ecosystem" [4] or "tumor as an organ" [5] explain ITH not as incidental phenomenon but as a

accession number "E-MTAB-9428" with the title "High accuracy gene expression profiling of sorted cell subpopulations from breast cancer PDX model tissue" (https://www.ebi.ac.uk/arrayexpress/experiments/E-MTAB-9428/).

**Funding:** This study was completely funded by BD Technologies and Innovation. The funder provided support in the form of salaries for authors WP, ES, FT, WSD, FH, MF and RB, but did not have any additional role in the study design, data collection and analysis, decision to publish, or preparation of the manuscript. The specific roles of these authors are articulated in the 'author contributions' section.

**Competing interests:** WP, ES, FT, WSD and RB are employees of BD Technologies and Innovation and FH and MF were employees of BD Technologies and Innovation at the time the work was carried out. The funder had no role in the study design, data collection and analysis, decision to publish, or preparation of the manuscript. The specific roles of these authors are articulated in the 'author contributions' section. This does not alter our adherence to PLOS ONE policies on sharing data and materials.

foundational property. Focusing on transcriptional heterogeneity in particular has become an effective way to link ITH to clinically meaningful phenomena [6]. Maximizing and understanding the accuracy of this approach will be an important factor in its efficient translation into the clinic.

scRNASeq has arguably been the main tool to accelerate discovery of transcriptional heterogeneity in tumors and healthy tissues alike. The results have prompted models which transcend the limits set by traditional cell type definitions and replace the artificially fixed categories with continuous gene expression spaces of individual cells [7, 8]. This is likely a much more accurate depiction of cell biological reality since each organism develops from a single cell in a continuous process rather than discrete "jumps". The drawback, however, lies mainly in a technical limitation: scRNASeq shows a 'dropout' effect due to inefficient mRNA capture [9] which makes it very difficult to create a high resolution model for a given sample. For elucidation of organisms (especially with high genetic homogeneity such as in inbred strains) this can be counteracted by sequencing large numbers of cells; maybe the most impressive example is represented in the "Tabula Muris", a high resolution analysis of 20 murine organs generated from over 100,000 cells [10]. This publication highlighted the enormous resources required but also showed that scRNASeq can be powerfully enhanced by combining it with cell sorting. Our investigations cause us to argue that the two methods are complementary rather than competing and demonstrate that this approach enables uncovering of highly reproducible gene expression patterns with striking functional implications.

Patient derived xenograft models allow propagation of human tumor tissue in immune compromised mice in ways that seem much superior to tissue culture conditions, particularly with regard to maintaining ITH [11]. Previously we had established methods in our lab that clearly demonstrate the presence of distinct subpopulations within breast cancer tissue emerging reproducibly in separate host animals implanted with tissue from the identical primary tumor [12]. Here we addressed the possible influence of fluorescence activated cell sorting (FACS) on transcriptional profiling and investigated the presence of multiple different subpopulations within the same breast cancer model tissue. As in the previous work we chose surface markers identifying cell compartments with sufficient numbers for molecular analysis of the subpopulations, for this model CD49f and CD133. Those two markers as well as CD184 which we had employed for the previous tumor model have been shown in numerous studies to allow identification of cancer cells with distinct roles in tumorigenesis and have frequently been used in the search of cancer stem cells [12–14]. Using quadruplicate samples we were able to demonstrate the potential this approach provides for detailed studies of specific cellular compartments within a complex tumor tissue. In addition, our data indicates that RNASeq of sorted PDX tumor cell populations complements scRNASeq through a) a much higher accuracy by avoiding the drop-out effects of single cell sequencing and b) providing a concept for replenishable sources of cell subpopulations with predictable gene expression signatures.

## Methods

All experimental procedures were essentially as previously described [12], any differences are noted below.

### PDX tumor extraction

The protocol was approved by the BD Institutional IACUC committee. BD is licensed by the USDA and is accredited by AALAC. All methods were carried out in accordance to best practices outlined in the *Guide for the care and use of Laboratory Animals* [15].

Mice were sourced from The Jackson Laboratory. The properties of the mouse model used in the study was a triple negative breast cancer model (BRC13, Jackson Lab designation TM00999) previously characterized for surface marker expression in our lab [12]. Female NOD/SCID/ILIIrg$^{-/-}$ (NSG) mice age 6 to 7 weeks were purchased from The Jackson Laboratory with tumor material already implanted. All animal studies were performed under an Institutional Animal Care and Use Committee–approved protocol. A dedicated room to safely house immune compromised mice was used. Positive air flow and filtration was maintained though a BioBubble system. Mice were socially housed in groups of five within a micro isolation cage rack. Food and water were available ad lib. Each cage contained environmental enrichments, including nestlets and igloos. Tumors were assessed weekly for growth and measured with a digital caliper. Tumor volume was estimated by the formula 0.5 x (length x width$^2$). Animals were monitored daily with additional measurements taken if needed. Animals were euthanized if grafted tumor tissue showed signs of necrosis. Additional health signs were monitored with the following termination points: "rapid" weight loss of >10%, specific clinical signs of severe organ involvement (e.g. circling, head-tilt, respiratory rates, paralysis). Also, Body Condition Score (BCS) was monitored, with a score of 2 or below serving as a humane endpoint. Once tumor size reached >500 mm$^3$ (several weeks to several months after inoculation), they were harvested for analysis. Mice were humanely euthanized via exsanguination under anesthesia at the time of tumor extraction, and the tumor was immediately removed via blunt dissection and placed in cold PBS for dissociation and analysis.

## Preparation of single-cell suspensions from tumor tissue

Single cell suspensions were prepared as previously described. Briefly, excised tissue was weighed and then minced using a scalpel or scissors to a size not exceeding 1 to 2 mm$^3$. Tissue was then enzymatically dissociated into single cells using Horizon™ Dri Tumor & Tissue Dissociation Reagent (BD Biosciences, San Jose, CA) in a 37°C water bath for 30 minutes with frequent agitation. The enzymatic reaction was stopped by rinsing in PBS (Cellgro) and 1% BSA (Sigma-Aldrich). Dissociated tissue was filtered through a 70 mm sieve and treated with ACK Buffer (Life Technologies) to remove contaminating red blood cells. Cells were resuspended in PBS, without magnesium and calcium.

## Cell staining for flow cytometry and analysis

Before staining, single-cell suspensions were blocked for 30 minutes, on ice in FcR block (Miltenyi Biotec,). For surface marker characterization, cells were diluted in PBS and distributed into wells of a 96-well plate containing viability dye (LIVE/ DEAD Fixable Near-IR or Aqua, Invitrogen) and Hoechst 33342 (Invitrogen; at 0.2 mg/mL) for viable nucleated cell identification. The basic sorting strategy was as described previously [12]. Monoclonal antibodies for CD49f (clone G0H3) were purchased from BD Biosciences, for CD133 from Miltenyi (Bergisch Gladbach, Germany). A complete list of reagents is provided in S1 Table. Cells were incubated with antibody solutions for 30 minutes at 4°C, in the dark, and then rinsed twice in PBS before acquisition on BD LSRII flow cytometer. For sorting, cells were stained using a four-color panel combining amine viability reagent, anti-mouse cocktail, and markers identified from the primary characterization plate. After staining, cells were rinsed, and resuspended in BD FACS Pre-Sort Buffer, filtered again through a 70 μ m sieve, and populations of interest were sorted on the BD FACSAria II (BD Biosciences) at 20 p.s.i. using a 100- μm nozzle. Cell doublets or clumps were removed with electronic doublet discrimination gating. Post-sorting analysis of sorted subpopulations regularly showed purity >95%. Analysis of results was performed using FACSDiva software (version 6.1.3 LSRll and version 8.0 FACSAria II). Graphical representations of the data were performed using R or MS Excel.

## RNA isolation, RNASeq and sequence data analysis

20–50,000 cells stained for a specific surface marker were collected during FACS sorting and processed in bulk for RNA extraction. Quality assessments were determined using Agilent BioAnalyzer and NanoDrop readings. RNA specimens underwent QC analysis to determine their suitability for target labeling. QC analyses included determination of concentration and volume, A260/A280 ratio, and 28S/18S rRNA ratio (where applicable), and an RNA integrity summary score (Agilent RIN or Caliper RQS). RNASeq was performed using the Illumina RNASeq TruSeq Stranded mRNA SQ905 Kit. All samples were sequenced in one run across four lanes on an Illumina HiSeq with 2 x 50 paired end reads. The primary sequencing analysis was done through a 3rd party (EA Genomics, Morrisville, NC) using STAR v2.4 for sequence alignment and RSEM v1.2.14 for transcription quantification.

RSEM counts from the sequencing results were used for the secondary analysis performed in R. Non-human genes were filtered out of the data set. Differential expression was calculated using the edgeR v3.14.0 package:

1.) Groups were created for each of the sorted populations ("Tissue", "Unsorted_0hr", "Unsorted_3hr", Human_Only", "CD49f_Low", CD49f_High", CD133_Low", CD133_High"). 2.) Normalization factors were calculated with calcNormFactors using the default settings, to convert observed library sizes to effective library sizes. 3.Counts per million (CPM) and log CPM were calculated from the normalized data. 4. glmQLFit was used to model the fit of the normalized data. 5. glmQLFTest was used to model pairwise comparisons between the groups. 6. topTags was used to extract the most differentially expressed genes from each comparison object created from step 5 (glmQLFTest). 7. The resulting differential expression data was used for comparison to 3rd party gene lists.

Statistical significance was calculated and displayed in all figures using false discovery rate (FDR).

To assess the biological interpretation of gene expression patterns all DEGs with FDR<0.1 and x-fold ≥2 were analyzed using the Enrichr tool [16, 17].

## Results

### Influence of tissue dissociation and FACS sorting on gene expression profiles

Our study was based on pairwise comparisons of gene expression profiles for (8) different cell populations from a triple negative breast cancer PDX model. The first (4) samples were used to validate our sample preparation and gene expression analysis processes, while subsequent samples were FACS sorted using (2) surface markers to compare biological similarities and differences in gene expression (Figs 1 and S1). Variables investigated in the sample preparation included tissue dissociation, duration of the method and flow sorting as possible sources of bias. RNA isolated from freshly collected tissue using a standard method served as a reference.

To control for the technical variability as well as biological variability between individual host mice, we carried out the experiment in quadruplicate, (i.e. each of the 8 conditions shown in Fig 1 represents 4 separate tumors from the same model grown in individual host animals harvested on different days) resulting in a total of 32 samples. Three samples did not pass QC due to limited amounts of RNA and/or poor RNA integrity.

Out of the 3 major variables (tissue dissociation, protocol duration and flow sorting) tissue dissociation was the only sample processing variable introducing significant changes in expression patterns (Fig 2). Among 24 differentially expressed genes (DEGs) after dissociation, 23 were upregulated above a cutoff of FDR<0.05/ 4-fold and only a single one was

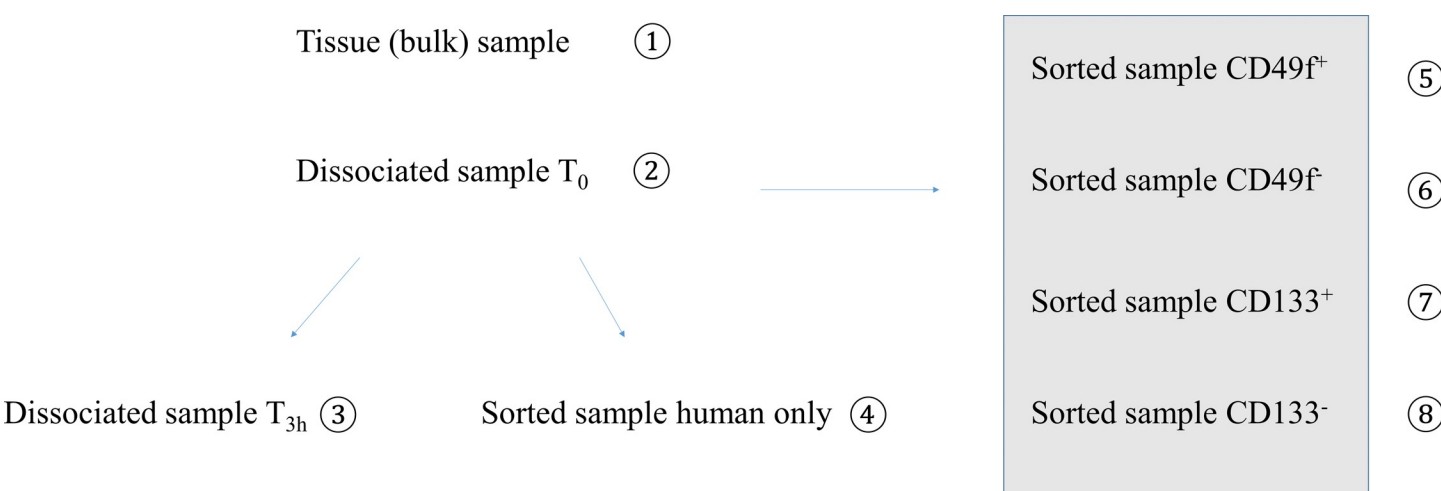

**Fig 1. Schematic of experimental setup.** Tumor tissue was extracted and gene expression profiles from 8 different samples per tumor were generated from 4 mice (i.e. total of 8x4 = 32 RNASeq profiles) of the same breast cancer PDX model, BRC13. The left part addressed the sample handling aspects of the procedure while the samples marked by the grey box were generated to investigate ITH present in this model. See text for details.

downregulated, suggesting *de novo* RNA synthesis rather than RNA degradation as the main effect triggered by this sample preparation step.

Investigation of the characteristics of the significantly changed genes implicated a specific mechanism causing these observations. A database query of the significantly changed genes

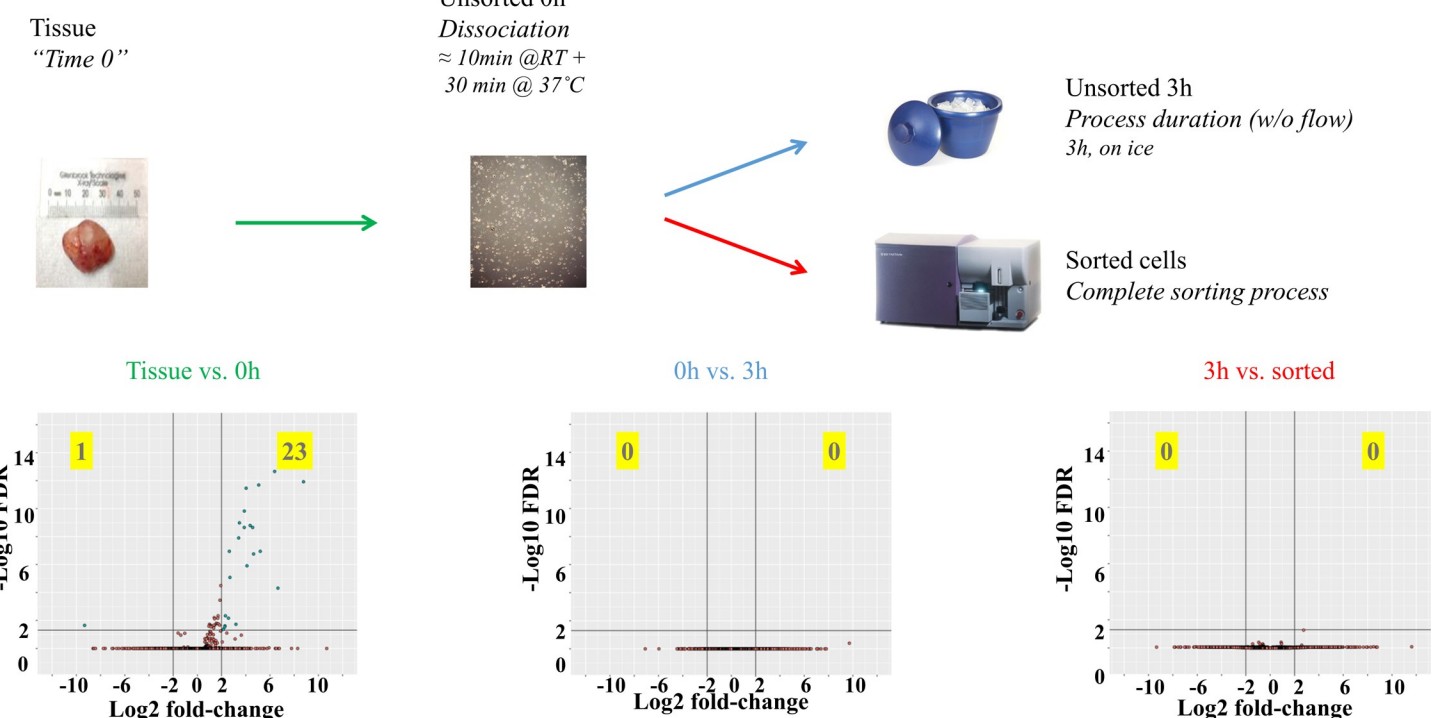

**Fig 2. Pairwise comparisons between gene expression profiles generated through bulk RNASeq for 4 different conditions.** 0h time point represents cells immediately after dissociation; 3h time point represents cells left undisturbed for 3 hours after dissociation while sorted cells underwent flow sorting which generally took up to 3 hours. Volcano plots show the log2 fold-change (x-axis) and the statistical significance as–log(FDR) (y-axis). Lines indicate the cutoff values chosen (4-fold and FDR<0.05). FDR values were calculated across the entire data set.

(Table 1) revealed that a great majority (20/24) have been described as immediate early genes (IEG). Of the 4 genes not identified in the IEG literature, 2 (AL832891 and AK024092) have 99% homology to other IEGs (Cyr61 and CHD2) and the remaining 2 are non-coding RNAs. Similar trends were seen with gene sets analyzed with lower stringency cutoff values (S2 Table).

## Gene expression profiles of cell subpopulations identified by CD49f and CD133

The second half of the experiments addressed ITH and whether subpopulations identified by surface markers differed in their gene expression profiles. Previously, we identified surface marker ITH in nine triple negative breast cancer models [12]. Each model had a unique and reproducible marker profile. Indication of ITH (i.e. presence of both positive and negative cells for a given marker) existed with respect to many of the markers but also differed among the models. Investigation of CD184^hi and CD184^lo sorted cells in one model revealed major differences in gene expression profiles between marker-high and low populations [12]. In the current study we focused on another PDX triple negative breast cancer model exhibiting varying expression levels of the surface markers CD49f and CD133. Both markers have been described as indicators of functionally distinct roles in cancer tissue such as drug resistance, metastasis, EMT and stemness [28–30]. For our investigation their choice was also opportunistic since they allowed isolation of sufficient cell numbers for both marker positive and negative

**Table 1. List of genes identified as DE before and immediately after tissue dissociation (FDR<0.05, x-fold>4).**

|     | Gene | FDR | x-fold change | Reference identifying gene as IEG |
| --- | --- | --- | --- | --- |
| 1 | FOS | 2.3E-15 | 263.0 | [18] |
| 2 | EGR1 | 2.2E-13 | 83.5 | [18] |
| 3 | FOSB | 1.2E-12 | 437.0 | [18] |
| 4 | DUSP1 | 2.0E-12 | 33.5 | [18] |
| 5 | IER2 | 3.5E-12 | 16.3 | [19] |
| 6 | JUN | 1.5E-10 | 14.7 | [18] |
| 7 | JUNB | 1.0E-09 | 11.1 | [18] |
| 8 | ZFP36 | 1.6E-09 | 20.6 | [18] |
| 9 | NR4A2 | 2.2E-09 | 23.7 | [18] |
| 10 | RHOB | 2.2E-09 | 14.7 | [20] |
| 11 | KLF2 | 1.2E-08 | 10.6 | [21] |
| 12 | GADD45B | 1.1E-07 | 6.2 | [18] |
| 13 | EGR2 | 1.1E-07 | 36.7 | [22] |
| 14 | NR4A1 | 1.8E-07 | 25.0 | [18] |
| 15 | ATF3 | 1.2E-06 | 17.1 | [18] |
| 16 | SOCS3 | 8.3E-06 | 6.5 | [23] |
| 17 | FAM71A | 4.9E-05 | 100.9 | [24] |
| 18 | AL832891 | 4.6E-03 | 5.0 | * |
| 19 | LOC284454 | 6.8E-03 | 5.9 | ncRNA |
| 20 | BTG2 | 1.9E-02 | 9.1 | [25] |
| 21 | CTD-2311B13.7:ENSG00000257931 | 2.2E-02 | -640.1 | ncRNA |
| 22 | CXCL2 | 2.4E-02 | 4.9 | [26] |
| 23 | AK024092 | 2.9E-02 | 4.8 | * |
| 24 | DUSP2 | 4.3E-02 | 4.4 | [27] |

* no reference found

phenotypes. The previous model as well as the one investigated here contained 15% of mouse cells which were excluded from the subsequent gene expression analysis [12].

We FACS sorted the dissociated tumor cell population into CD49f[hi] and CD49f[lo] as well as CD133[hi] and CD133[lo], and performed (whole transcriptome) RNASeq on each pool. Both sorting approaches identified a number of DEGs, even at very stringent statistical significance levels (Fig 3 and Table 2), suggesting significant differences between the respective subpopulations. The finding that gene expression correlated with protein expression (i.e. ITGA6 for CD49f and PROM1 for CD133 were significantly upregulated in the marker positive population) suggests gene variance is due to biological changes and not due technical noise. Moreover, the statistical significance for both genes was highest among upregulated genes in the marker positive population in both experiments (S2 Fig).

Among all genes identified in both experiments the majority were unique to one arm but among the ones in common (ca. 1/3 of the CD133 gene set overlapped with 1/8 of the CD49f set) an unexpected inverse correlation was detected (Fig 4). 120 of the 121 genes in common were inversely correlated at a qualitative level (i.e. identical direction of change) and at a quantitative level the correlation coefficient was R = - 0.85, strongly suggesting that this was not caused by random noise.

A priori an inverse correlation could be a sign of a simple negative correlation between the two markers. However both the flow analysis (Fig 4A) as well as the finding that more DE genes are unique to one arm of the experiment than common in both arms argue for a more complex relationship between the populations.

To investigate the possible roles for the respective subpopulations we determined GO terms for each of the 6 upregulated gene sets (S5–S10 Tables). Possibly the best defined functional indicator could be determined for the BRC13/CD133[hi] population which–very much like the BRC12/CD184[hi] population contained many GO terms associated with cell proliferation (S6 and S10

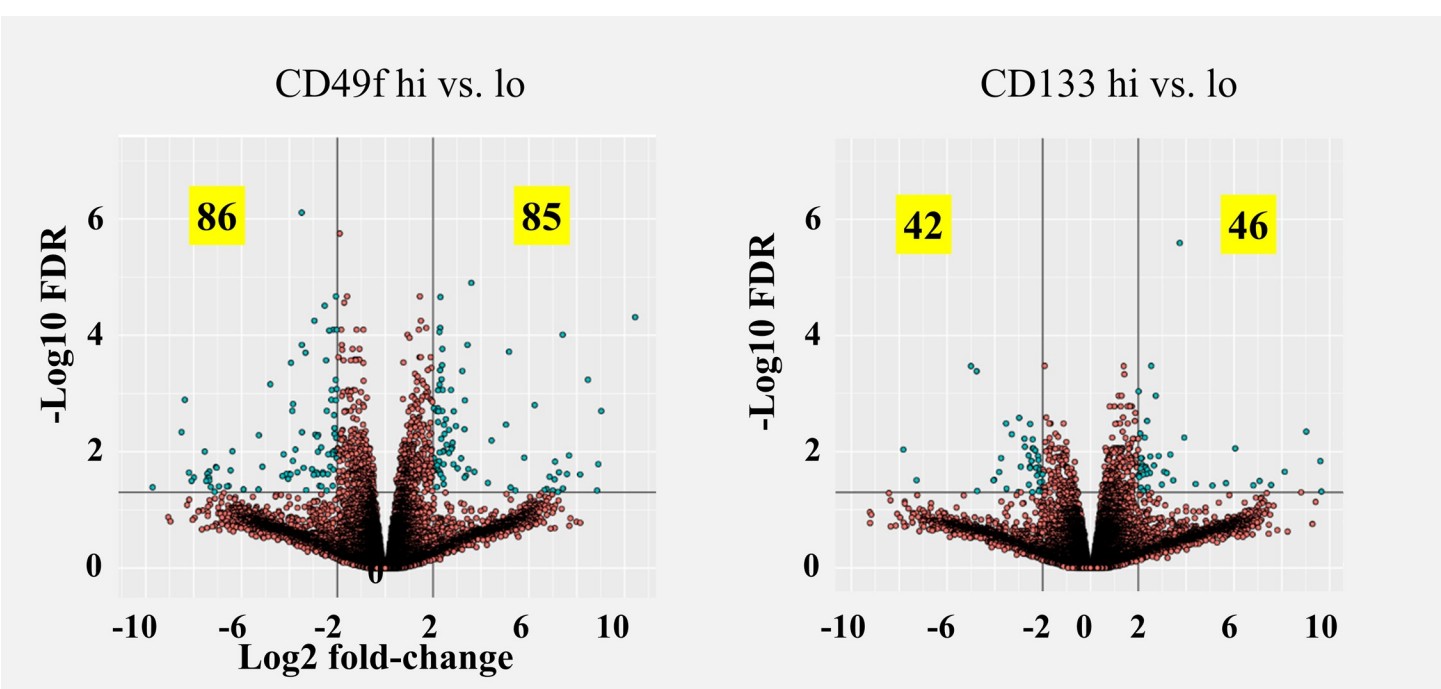

**Fig 3. Volcano plots of DEGs identified between tumor subpopulations sorted with either CD49f (left) or CD133 (right).** Cutoff values shown (black lines, for illustration only) are >4-fold and FDR <0.05.

**Table 2. Number of differentially expressed genes at different cutoff levels.**

| Cutoff | CD49f<sup>hi</sup> | CD49f<sup>lo</sup> | CD133<sup>hi</sup> | CD133<sup>lo</sup> |
|---|---|---|---|---|
| 0.1 | 668 | 620 | 432 | 274 |
| 0.05 | 423 | 369 | 233 | 161 |
| 0.01 | 150 | 129 | 57 | 33 |

Tables). The BRC13/CD49<sup>lo</sup> population shows a variety of signs consistent with organization of the microenvironment, such as ECM organization, angiogenesis and regulation of cell migration which all appear as cellular processes indicated for the BRC12/CD184<sup>lo</sup> subpopulation (S7 and S9 Tables). The genes upregulated in BRC13/CD49f<sup>hi</sup> and BRC13/CD133<sup>lo</sup> populations did not provide GO terms which would indicate a clear consistent function but–consistent with the inverse correlation described above- 8 out of the top 50 GO terms in those two lists are identical (S5 and S8 Tables). While neither BRC13 DEG list provided evidence for a strong and unambiguous functional interpretation of the gene profiles, the CD133<sup>hi</sup> and CD49f<sup>lo</sup> lists mirror a number of gene sets identified in breast cancer as well as other solid cancers, indicating the possibility of functional relevance of the subpopulations described. Particularly, the BRC12/CD184<sup>lo</sup> and BRC13/CD49f<sup>lo</sup> populations contained many overexpressed genes in common with profiles described under conditions of hypoxia (Fig 5A, addt'l examples in supplementary data). In contrast, the BRC12/CD184<sup>hi</sup> and BRC13/CD133<sup>hi</sup> populations contained equally pronounced similarities to profiles identified in connection with high proliferation in healthy breast tissue, regulation by the E2F transcription

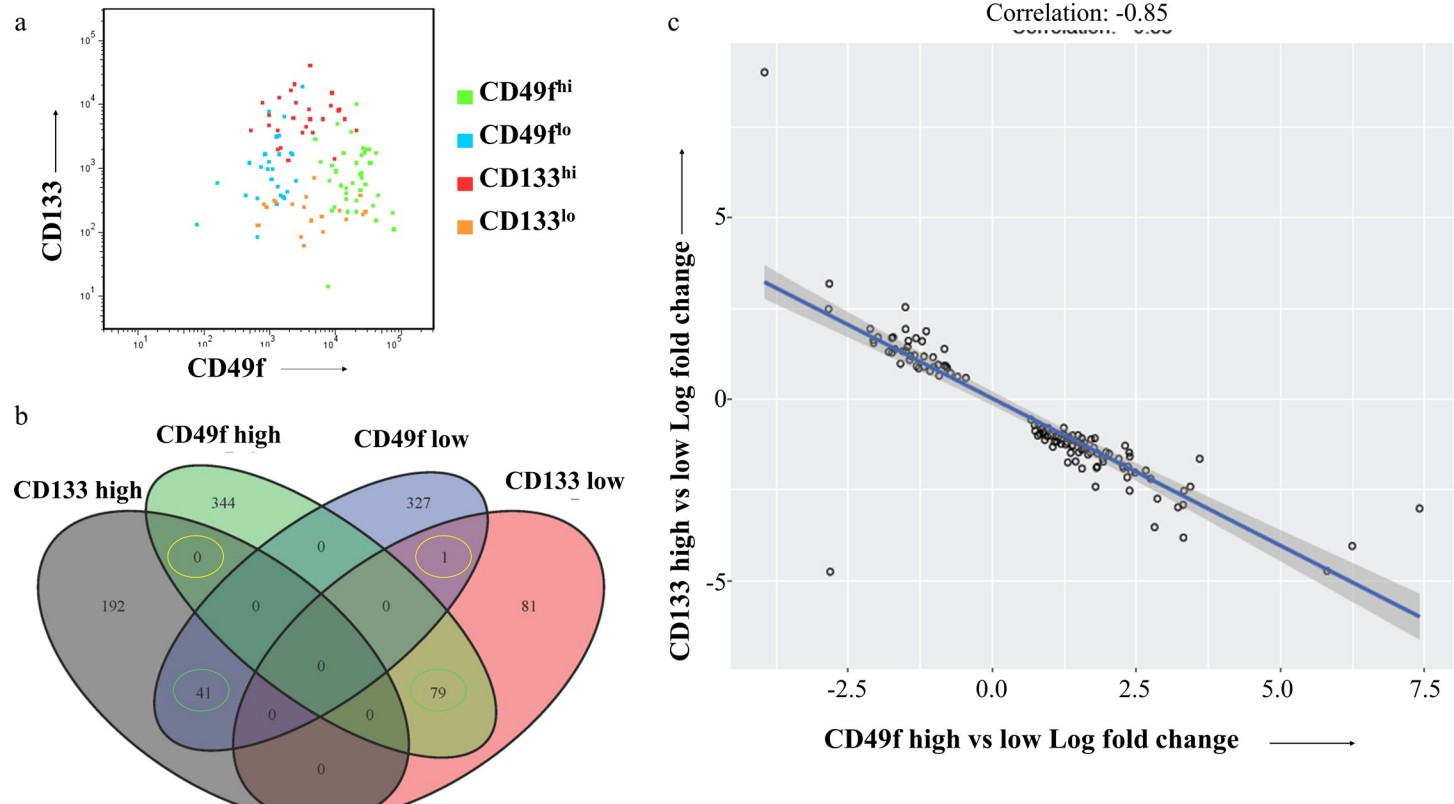

**Fig 4.** Flow cytometric analysis of sorted populations using CD49f and CD133 (a) and correlations between DEGs identified by using either CD49f or CD133 as sorting criteria (b and c, FDR<0.05).

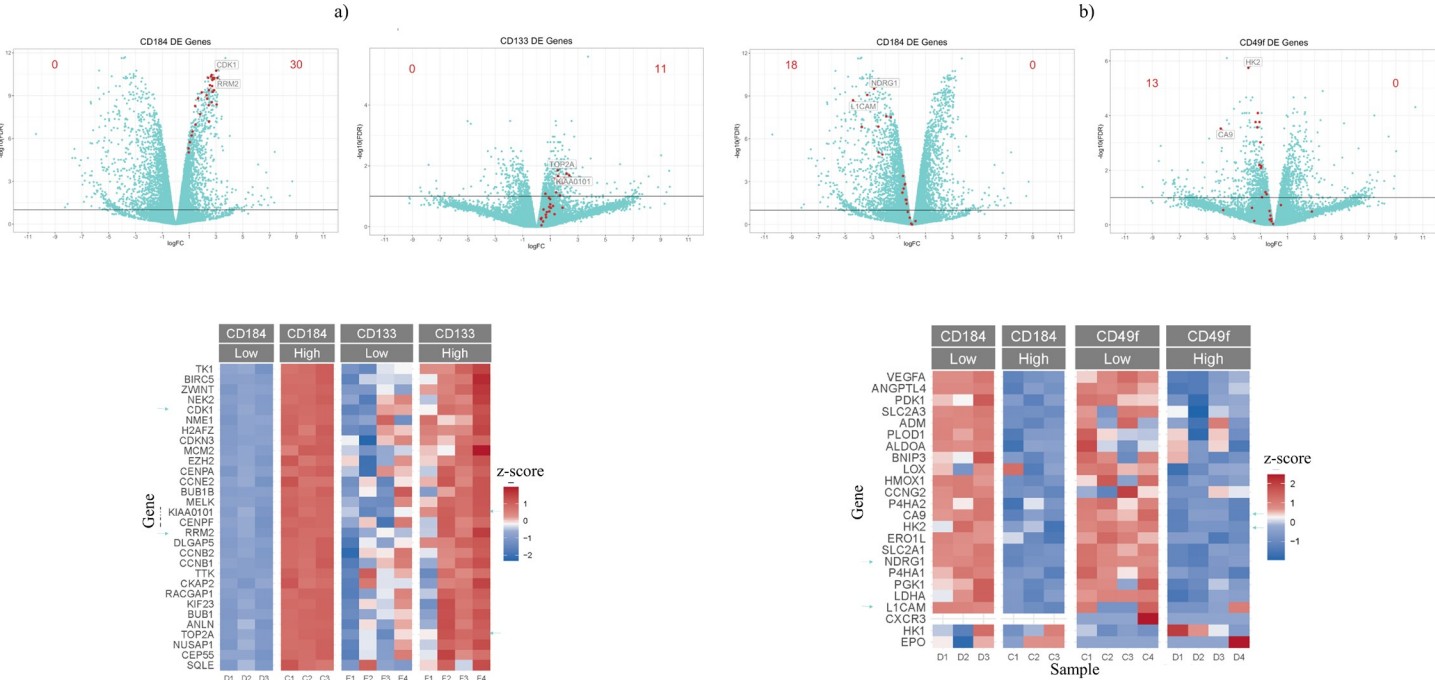

**Fig 5. Gene expression signatures identified in BRC12 and BRC13 tumor cell subpopulations.** A hypoxia gene signature composed of 3 published signatures [30–32] was found in BRC12 CD184[lo] and BRC13 CD49f[lo] subpopulations (a) while a proliferative gene signature identified in histologically normal breast tissue from cancer patients [41] was present in BRC12 CD184[hi] and BRC13 CD133[hi] subpopulations (b). Genes with lowest FDR and highest x-fold values are labeled in volcano plots and marked with arrows in heatmaps, respectively.

factor family, DNA damage response and stem cell properties (Fig 5B and supplementary data). Such parallels to previously documented results could also be identified within functional gene families; especially within the keratin family of genes 8 out of 9 members DE with FDR<0.1 in both BRC12/CD184[lo] and BRC13/CD49f[lo] cells tracked in almost identical patterns (S3 Table).

## Discussion

Tissue transcriptional profiling is dependent on the accurate reflection of cells in their physiological environment. In other words, distortion caused by sample handling or signal amplification are a frequent hurdle to identifying genes that are consistently expressed in a differential state that reflects physiology [31]. We established a workflow that accurately identifies distinct subpopulations in tissue from breast cancer PDX models [12]. In this study, we set out to determine if sample handling or flow cytometry introduced bias in transcriptional profiling and show different subpopulation profiles indicating complex ITH in tumor tissue from a breast cancer PDX model. The surface markers were chosen based on their expression levels within this model rather than for known biological roles.

To test accuracy we compared expression profiles assessed by whole genome RNASeq from dissociated tissue cells before and after FACS sorting with the "gold standard" profile of bulk fresh tissue. Using quadruplicate samples (i.e. 4 separate tumors from the same tumor model) allowed us to statistically assess random fluctuations in individual samples from consistent gene expression differences caused by the steps investigated. We found no evidence of compromised mRNA integrity in any of the steps including flow sorting. Our results indicate that tissue dissociation caused significant changes in a limited set of genes. However, outside of this specific group (IEGs representing less than 0.5% of the human genome) no significant

expression differences were detected when compared to fresh tissue. In contrast, transcriptional profiling of sorted subpopulations derived from this tissue identified distinct gene clusters differentially expressed in cells identified by specific surface markers. Furthermore, our data provides evidence that those differences can be detected with extremely high precision.

IEGs have been extensively described in the literature as fast responders to multiple stimuli. It's been shown that IEGs in a variety of cell types are activated within minutes after exposure to various perturbations, including mechanical stress [32]. Their ability to rapidly activate transcription likely represents part of a cascade of events that are involved in tight gene regulation. The cascade is initiated with a distinct set of immediate early genes, followed by a set of delayed early genes [18, 33], a nomenclature derived from the sequence of viral gene expression after successful entry of the virus into its host cell. Due to this cascade of early events, it's unlikely any published tissue dissociation protocol (or detachment of adherent tissue culture cells) will prevent activation of IEGs. However, our data suggests artefactual changes to the rest of the transcriptome can be avoided, presumably by limiting the ability of cells to execute the full cascade. The specific list of IEGs involved in a given experiment is context dependent [32]. Our survey of published literature revealed that many IEGs overlap between different experiments and their total number is unlikely to exceed 150–200 genes. As a consequence, we have shown that our protocol allows us to effectively limit process related bias and accurately detect gene expression changes in >99% of genes. Bioinformatic filtering of all known IEGs and inclusion of control samples offer additional measures to mitigate artefacts when processing cells derived from tissue. It is also remarkable that aside from the IEGs discussed above, fewer than 200 genes would be categorized as DE even up to a FDR <0.9 cutoff. This strongly argues that no systematic distortion is caused by the methods used here and experimental noise is minimal (see S3 Fig). In conclusion, our data suggests that our methods detect DEGs between different experimental conditions with high accuracy and reproducibility.

The exploratory segment of our experiment targeted the discovery of gene expression profiles of interest in tumor tissue subpopulations identified by various surface markers. A previously published example from our lab had validated this approach [12] and we aimed to uncover whether using different surface markers as well as a different tumor model may result in similar or different subpopulation profiles.

We successfully identified distinct DEG sets with each marker containing hundreds of DEGs even at stringent statistical cutoff values (Table 2). Several arguments derived from our results would indicate that RNASeq performed on FACS sorted cell populations enables sensitive and precise identification of genes unique to these cells: 1) The genes for both surface markers used in this study (ITGA6/CD49f and Prom1/CD133) as well as our previous study (CXCR4/CD184) were among the top 3 DEGs (ranked by FDR, S2 Fig). 2) Among the range of differentially expressed genes with very high statistical significance (FDR<0.01) a number were detected at lower than 2-fold differential expression (8 for CD133 and 39 for CD49f sorting). 3) The DEGs common to both CD133 sorting as well as CD49f sorting were strongly correlated; the 121 genes in common between each sort (at FDR<0.05) have a coefficient of R = -0.85 (Fig 4). Such a clear correlation is much more consistent with an underlying biological control of those genes as opposed to a random occurrence and further supports the notion of a very low noise level. All of these observations are consistent with "clean" separation of expression profiles in subpopulations isolated through FACS sorting.

When we compared the two DEG lists from this work with our previously published list (using a different TNBC model labeled BRC12) and with gene profiles from other investigators we found a number of surprising parallels. For the two cancer models used in our lab there was substantial overlap between DEGs in the BRC12/CD184[lo] and BRC13/CD49f[do] populations on one hand and BRC12/CD184[hi] and BRC13/CD133[hi] on the other (Figs 5 and S4–

S13). This was not only far more than would be expected by chance but each overlap also aligned with gene expression patterns described by other investigators related to biological functions important to tumorigenesis. BRC12/CD184[lo] and BRC13/CD49f[lo] cells expressed many genes related to hypoxia and drug resistance [34–36], (Figs 5A and S10–S13) while BRC12/CD184[hi] and BRC/CD133[hi] cells had high expression in genes indicating proliferation, DNA Damage Response and stemness (Figs 5B and S4–S9).

Focusing on specific gene families we found further evidence of co-regulation of genes in a manner consistent with functional relevance. The best such example is presented in the keratin gene family. The KRT genes 5,6, 14, 16 and 17 which appear to be co-regulated and have been used as markers for basal breast cancer [37–40] are all upregulated in BRC12/CD184[lo] and BRC13/CD49f[lo] cells with remarkable quantitative consistency (S3 Table). Their upregulation is also consistent with the activity of Hypoxia Inducible Factor (HIF), i.e. the presence of hypoxia [41, 42]. Furthermore, especially some genes for which evidence for functional roles in hypoxia has been demonstrated repeatedly and in cancer cells (e.g. L1CAM, ANGPTL4, CA9, VEGFA, LDHA, P4HA1, LOX, LOXL2, LOXL4) are almost all significantly upregulated in both BRC12/CD184[lo] and BRC13/CD49f[lo] cells (S3 Table). Hypoxia is often correlated with therapeutic resistance [43, 44]. In accordance with this, gene expression signatures of drug resistance can also be found in those same populations (S8 Fig).

While functional tests clearly are required to confirm phenotypic relevance of our results we believe that these distinct distributions of overexpressed genes in separate compartments point to some very intriguing possibilities. The two mouse models we investigated showed a high degree of similarities, especially considering the functions the gene patterns are implicated in. scRNASeq has led to a much better appreciation of subtle gene expression differences between otherwise similar tissue cells and a number of those studies with TNBC result in similar gene patterns as we describe here (S9–S11 Figs). Often gradients are found across different locations within the tissue but they don't necessarily exist across straight lines which is one among multiple challenges when investigating their functional relevance. It may be useful to expand on the historical notion of what a "cell type" is by considering "cell states" with distinct functional properties [45]. In a recent publication Uri Alon and colleagues proposed a model where cells exist along different axes within a multidimensional space of functionality and each end of a given axis is defined by the presence of "specialist" cells with respect to the function in question [8]. scRNASeq is highly efficient at detecting such gradients but gene dropout effects limit its resolution and function can only be deduced from correlations. A variety of spatial methods have been developed to locate cells and reveal the tissue architecture but similar limitations apply and those studies are highly resource intensive. We believe our approach is complementing those tools by providing direct access to such "specialist" cells in quantities sufficient to carry out in depth functional investigations. Acquired cancer drug resistance might be a primary example of where this could be of major clinical use. Together with the findings of other investigators our data argue that such "division of labor" exists in breast cancer tissue where hypoxia/drug resistance may reside in one compartment while proliferation/ stemness/DNA damage response may be driven by another. We show for the first time that such compartments emerge spontaneously and reproducibly in different PDX models. Some overlaps of our profiles with drug resistance signatures in other cancer types (e.g. lung, melanoma; see S8 Fig) suggests that this type of ITH may be common across solid cancer tissues. This provides powerful options for setting up prospective studies in vivo. Recent developments of PDX models with humanized immune systems [46] will likely further improve those options. Studies targeting drug response variations have been conducted in cell culture [44] and resulted in changes of many of the genes related to hypoxia and TGFβ activity described in our study.

A number of studies have described ITH regarding both phenotype as well as genotype and suggested interactions between the different compartments [47–49]. The successive engraftment of heterogeneous patient derived cancer tissue through multiple passages [47] showed continued co-existence of distinct genotypic clones growing in each new generation of host mice. The implausibility of perfectly synchronized independent genotypes suggests that clones are interdependent which is consistent with numerous reports of non-cell autonomous proliferation [50, 51]. Our data do not include DNA sequencing and it is thus unclear whether the surface marker identified subpopulations are genotypically distinct, identical or mixed themselves. The overlapping gene expression patterns we observed with the different subpopulations identified in BRC13 (Fig 4) are more consistent with a model of gradients within multidimensional gene expression space as the one pioneered by Uri Alon and colleagues [7] than with a mixture of uniform clones. In our experience surface marker profiles across different mice from the same model, including BRC12 and BRC13 remain quite stable [12]. Although this could be interpreted as a sign of stable relative ratios but it is important to keep in mind that for technical reasons most PDX tissue is analyzed within a fairly narrow range of tumor sizes. Some investigators reported observations that both the relative clonal ratios and total combined cell numbers influence growth kinetics [48] and it is certainly possible that the relative ratios might differ if the same tumors are harvested at a different stage of their growth. Studies combining our approach with DNA sequencing, lineage barcoding and propagation of clonal mixtures through multiple passages as well as analysis at different stages of tumor development provide many opportunities to clarify the respective contribution of each category of ITH described. One of the two PDX models investigated by Jeff Chuang and colleagues [47] happens to be the same one we used in our previous study (BRC12 aka TM00096) and thus would offer a particularly rich data set as background for further work. The use of short-term cultures of defined mixtures of cells as described by Bruna et al. [49] offers an additional efficient way towards such investigations. We believe our work describes a near optimal source of cells for this purpose since reproducibly emerging subpopulations with well defined gene expression patterns grown in the same tumor eliminate the confounding factors afflicting cells grown on plastic or derived from multiple patients. The two most important implications of our findings are technical and biological in nature, respectively: 1) our approach is an efficient option to investigate subpopulations at a functional level due to their accessibility through well established methods of surface marker directed FACS sorting. 2) subpopulations with distinct gene expression signatures related to hypoxia/drug response and proliferation/DNA damage response may be common across different breast cancer models prior to drug challenges. We have successfully applied FACS analysis based on these methods to patient derived tissues [52] raising the possibility that once the functional implications of the various gene profiles can be better understood optimized assays targeted to clinical samples could likely be developed.

## Supporting information

**S1 File.**
(DOCX)

**S1 Fig. Flow cytometric analysis of representative BRC 13 sample.** Plots represent sample before (left) and after (right, overlay of 4 samples) sorting with CD49f and CD133, resp. These plots are representative of all replicate tumors.
(PDF)

**S2 Fig. Surface markers used for sorting and respective gene expressions.** Genes encoding the surface markers which were used to sort the populations (CD184/CXCR4, CD49f/ITGA6,

CD133/Prom1) were found to be differentially expressed with high statistical relevance in each experiment. CD184 (left) data is from a prior publication [12]. The horizontal line marks FDR = 0.1.
(PDF)

**S3 Fig. Number of genes on the y-axis shown as a function of the FDR.** Even up to very high FDR values the number of genes assessed to be different is very low for all comparisons between samples during the sample prep phase while the number of genes differentially expressed between sorted subpopulations grows continuously with increasing FDR.
(PDF)

**S4 Fig. Gene expression signatures identified in BRC12 and BRC13 tumor cell subpopulations for breast tissue proliferation.** Overlap of a proliferative gene signature from histologically normal breast tissue (30 genes, [53], top; same as in Fig 5) and "hallmark E2F target genes" (34 genes, [54], bottom) with resp. gene sets within this publication are shown in volcano plots Numbers provided are genes above FDR cutoff of 0.1 up (right) or down (left) regulated.
(PDF)

**S5 Fig. Gene expression signatures identified in BRC12 and BRC13 tumor cell subpopulations for breast tissue proliferation.** Overlap of a proliferative gene signature from histologically normal breast tissue (30 genes, [53], left; same as in Fig 5) and "hallmark E2F target genes" (34 genes, [54], right) with resp. gene sets within this publication are shown in heatmaps. 2 genes (TOP2A and MCM2, see arrows) overlap between the signatures.
(PDF)

**S6 Fig. Gene expression signatures identified in BRC12 and BRC13 tumor cell subpopulations for stemness and genomic instability.** Overlap of a stem cell gene signature [55], (top/) and a genomic instability signature identified in glioma [56] (bottom) are shown in volcano plots.
(PDF)

**S7 Fig. Gene expression signatures identified in BRC12 and BRC13 tumor cell subpopulations for stemness and genomic instability.** Overlap of a stem cell gene signature [55], (left) and a genomic instability signature identified in glioma [56] (right) are shown in heatmaps (b).
(PDF)

**S8 Fig. Gene expression signatures identified in BRC12 and BRC13 tumor cell subpopulations for poor outcome and DNA damage response.** Overlap of a subpopulation signature associated with poor outcome in lung adenocarcinoma [57] (top) and a signature associated with DNA damage response (DDR) and resistance to PARP inhibition [58] (bottom) are shown in volcano plots.
(PDF)

**S9 Fig. Gene expression signatures identified in BRC12 and BRC13 tumor cell subpopulations for poor outcome and DNA damage response.** Overlap of a subpopulation signature associated with poor outcome in lung adenocarcinoma [57] (left) and a signature associated with DNA damage response (DDR) and resistance to PARP inhibition [58] (right) are shown in heatmaps (b).
(PDF)

**S10 Fig. Gene expression signatures identified in BRC12 and BRC13 tumor cell subpopulations for hypoxia.** Overlap of a hypoxia signature composed from 3 publications (see Fig 5a, [34–

36], top) and a larger hypoxia gene set described in [59], (bottom) are shown in volcano plots.
(PDF)

**S11 Fig. Gene expression signatures identified in BRC12 and BRC13 tumor cell subpopulations for hypoxia.** Overlap of a hypoxia signature composed from 3 publications (see Fig 5a, [34–36], left) and a larger hypoxia gene set described in [59], (right) are shown in heatmaps (b).
(PDF)

**S12 Fig. Gene expression signatures identified in BRC12 and BRC13 tumor cell subpopulations for drug resistance.** Overlap of a drug resistance signature described in lung carcinoma and melanoma [60] are shown in a volcano plot.
(PDF)

**S13 Fig. Gene expression signatures identified in BRC12 and BRC13 tumor cell subpopulations for drug resistance.** Overlap of a drug resistance signature described in lung carcinoma and melanoma [60] are shown in a heatmap.
(PDF)

**S14 Fig. Gene expression signatures identified in BRC12 and BRC13 tumor cell subpopulations with overlap of clusters identified in scRNASeq experiments on TNBC samples.** Overlap of 19 genes associated with proliferation in a sscRNASeq analysis of TNBC tissue [61].
(PDF)

**S15 Fig. Gene expression signatures identified in BRC12 and BRC13 tumor cell subpopulations with overlap of clusters identified in scRNASeq experiments on TNBC samples.** Overlap of 18 genes associated with an epithelial phenotype in a scRNASeq study on TNBC patient samples [62].
(PDF)

**S16 Fig. Gene expression signatures identified in BRC12 and BRC13 tumor cell subpopulations with overlap of clusters identified in scRNASeq experiments on TNBC samples.** Overlap genes identified in different clusters within single cells from the same tissue in PDX models of TNBC [63] A) 24 combined genes most upregulated in clusters 1,3,6 B) 20 combined genes most upregulated in clusters 2,4.
(PDF)

**S1 Table. List of reagents used in flow sorting.**
(XLSX)

**S2 Table. IEGs identified with statistical cutoff of FDR<0.1.**
(XLSX)

**S3 Table. List of GO terms generated in Enrichr for DEGs upregulated in the BRC13/ CD133<sup>lo</sup> subpopulation.**
(XLSX)

**S4 Table. List of GO terms generated in Enrichr for DEGs upregulated in the BRC13/ CD133<sup>hi</sup> subpopulation.**
(XLSX)

**S5 Table. List of GO terms generated in Enrichr for DEGs upregulated in the BRC13/ CD49f<sup>lo</sup> subpopulation.**
(XLSX)

**S6 Table. List of GO terms generated in Enrichr for DEGs upregulated in the BRC13/CD49f^hi subpopulation.**
(XLSX)

**S7 Table. List of GO terms generated in Enrichr for DEGs upregulated in the BRC12/CD184^lo subpopulation.**
(XLSX)

**S8 Table. List of GO terms generated in Enrichr for DEGs upregulated in the BRC12/CD184^hi subpopulation.**
(XLSX)

**S9 Table. Select genes differentially expressed in BRC12/CD184^lo and BRC13/CD49f^do.** A number of members of the keratin family of genes (a) as well as functionally relevant hypoxia genes (b) were found to be regulated in a very similar fashion in BRC12 and BRC13 subpopulations.
(XLSX)

**S10 Table. Gene lists with expression profiles selected from literature.**
(XLSX)

**S11 Table. DEG list from previous experiment using BRC12 and CD184.**
(XLSX)

**S12 Table. DEG list from 2^nd experiment using BRC13 and CD49f.**
(XLSX)

**S13 Table. DEG list from 2^nd experiment using BRC13 and CD133.**
(XLSX)

## Acknowledgments

The authors wish to thank Drs. Jeff Baker, Eric Dixon, BDTI and Ben Stanger, University of Pennsylvania for helpful comments.

## Author Contributions

**Conceptualization:** Warren Porter, Eileen Snowden, W. Shannon Dillmore, Rainer Blaesius.

**Data curation:** Warren Porter, Frances Tong.

**Formal analysis:** Warren Porter, Eileen Snowden, Frances Tong, Rainer Blaesius.

**Investigation:** Warren Porter, Eileen Snowden, Friedrich Hahn, Mitchell Ferguson, Rainer Blaesius.

**Methodology:** Warren Porter, Eileen Snowden, Friedrich Hahn, Mitchell Ferguson, Rainer Blaesius.

**Project administration:** W. Shannon Dillmore, Rainer Blaesius.

**Software:** Warren Porter.

**Supervision:** W. Shannon Dillmore, Rainer Blaesius.

**Validation:** Warren Porter, Eileen Snowden, Frances Tong.

**Visualization:** Warren Porter, Eileen Snowden, Frances Tong, Rainer Blaesius.

Writing – **original draft:** Rainer Blaesius.

Writing – **review & editing:** Rainer Blaesius.

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
