## [Decision Letter · Decision Letter 0]

15 Jun 2020

PONE-D-20-11376

High accuracy gene expression profiling of sorted cell subpopulations from breast cancer PDX model tissue

PLOS ONE

Dear Dr. Blaesius,

Thank you for submitting your manuscript to PLOS ONE. After careful consideration, we feel that it has merit but does not fully meet PLOS ONE’s publication criteria as it currently stands. Therefore, we invite you to submit a revised version of the manuscript that addresses the points raised during the review process.

Thank you for your patience with the review process. We apologize for the length of time required durign the COVID-19 pandemic. 

We look forward to receiving your revised manuscript.

Kind regards,

Randall J. Kimple

Academic Editor

PLOS ONE

Journal Requirements:

2. At this time, we request that you  please report additional details in your Methods section regarding animal care, as per our editorial guidelines:

(1) Please provide details of animal welfare (e.g., shelter, food, water, environmental enrichment)

(2) Please include the method of euthanasia  

(3) Please describe the care received by the animals, including the frequency of monitoring and the criteria used to assess animal health and well-being during the course of the tumor study.

Thank you for your attention to these requests.

3. Thank you for including your ethics statement: 'All animal studies were performed under an Institutional Animal Care and Use Committee–approved protocol.'  

(a) Please amend your current ethics statement to include the full name of the ethics committee/institutional review board(s) that approved your specific study.  

(b) Once you have amended this/these statement(s) in the Methods section of the manuscript, please add the same text to the “Ethics Statement” field of the submission form (via “Edit Submission”).

4. Please note that PLOS does not permit references to “data not shown.” Authors should provide the relevant data within the manuscript, the Supporting Information files, or in a public repository. If the data are not a core part of the research study being presented, we ask that authors remove any references to these data.

5. Please provide the product number and any lot numbers of the primary antibodies purchased from chemical companies for your study.

6. To comply with PLOS ONE submission guidelines, in your Methods section, please provide additional information regarding your statistical analyses. For more information on PLOS ONE's expectations for statistical reporting, please see https://journals.plos.org/plosone/s/submission-guidelines.#loc-statistical-reporting.

7. We note that you have indicated that data from this study are available upon request. PLOS only allows data to be available upon request if there are legal or ethical restrictions on sharing data publicly. For more information on unacceptable data access restrictions, please see http://journals.plos.org/plosone/s/data-availability#loc-unacceptable-data-access-restrictions.

8. Thank you for stating the following in the Competing Interests section:

WP, ES, FT, WSD and RB are employees of BD and FH and MF were BD employees

at the time the work was carried out.

We note that one or more of the authors are employed by a commercial company: BD Technologies and Innovation,

Reviewers' comments:

Reviewer's Responses to Questions

**Comments to the Author**

1. Is the manuscript technically sound, and do the data support the conclusions?

Reviewer #1: Partly

Reviewer #2: Yes

2. Has the statistical analysis been performed appropriately and rigorously? 

Reviewer #1: I Don't Know

Reviewer #2: Yes

3. Have the authors made all data underlying the findings in their manuscript fully available?

Reviewer #1: Yes

Reviewer #2: Yes

4. Is the manuscript presented in an intelligible fashion and written in standard English?

Reviewer #1: Yes

Reviewer #2: Yes

5. Review Comments to the Author

Reviewer #1: In “High accuracy gene expression profiling of sorted cell subpopulations from breast cancer PDX model tissue”, Porter et al study the influence of fluorescence activated cell sorting (FACS) on transcriptional profiling and investigate the presence of multiple different subpopulations within the same breast cancer model tissue.” The manuscript seeks to understand whether there are consistent expression features of cell subpopulations in tumors, by using a xenograft model system. This is an important question for the field. While the techniques used are not at the forefront (scRNA-seq would be preferable), the concept of the work is sound. I have a few major concerns and some minor critiques.

Major

Line 180: The authors should provide a formal gene set analysis of the genes that were DE in both experiments, as well as for the DE genes unique to individual experiments. The analysis provided in lines 192-203 is relevant to this but is too inexact to be evaluated.

The authors have a sizable number of RNA-seq replicates (line 132 suggests there are 32 RNA-seq datasets, though it is difficult to tell from the text) both across and within xenografts. However, there is little discussion of the robustness of the DE analysis across and within mice. This high number of replicates is the major strength of the work, so comparisons of replicates should be explicit.

The authors should describe how many cells were in each of the sorted populations. The proportions of cell populations that appear within xenograft ecosystems has been studied in pubs such as [Cell 167, 260–274.e22 (2016); Nat Comms 5, 5871 (2014); Sci Rep 8, 17937 (2018)] and may be relevant for tumor growth. Comments on this topic would strengthen the work. On a related point, are there published scRNA TNBC datasets available that the authors can compare to? Comparisons to such data would clarify the robustness of the observed effects.

Minor

Line 146: Please clarify that the references indicate the genes’ IEG status. This is not apparent from the table 1 caption.

Line 182: “120 of the 121 genes in common were qualitatively correlated (i.e. identical direction of change) and on a quantitative level the correlation coefficient was R= - 0.85” : The first and second parts of the sentence are stated inconsistently. If the direction of change is identical then the correlation coefficient should be positive. Rephrase.

Some aspects of the article are out of date. E.g. Ref 12 lists a publication from 2017 as “in press.”

Reviewer #2: Porter et al. have submitted a will written report investigating intratumoral heterogeneity of the transcriptome within solid human tumors by isolation of distinct cancer cell populations by FACS. Across tumor cases and replicates the authors identified immediate early gene changes, primarily due to tumor dissociation, that agreed to a high degree with previous publications. Cell populations were sorted based on high/low expression of the cell surface markers CD49f, CD133, and CD184, which were then analyzed by RNAseq. Differentially expressed genes were identified between populations indicating that each population may have distinct roles within the tumors. Overall, the experiments were designed well with proper controls, and this research will likely pave the way for more functional assays in the future (e.g. therapeutic response). I have no major concerns with the manuscript and have only a few minor comments/suggestions.

- In the Introduction I would suggest adding a brief description of the biological roles of CD49f, CD133, and CD184 and/or the types of cancer cell populations they have previously been used to isolate and define.

- For whole fresh tumor and the “bulk” dissociated cell population, what was the degree to mouse stromal (fibroblast, endothelial, and immune cell) cell infiltration? This can often be 10-50% in solid tumors. Did this have any effect on the RNAseq analysis for these samples?

6. PLOS authors have the option to publish the peer review history of their article (what does this mean?). If published, this will include your full peer review and any attached files.

Reviewer #1: No

Reviewer #2: No

---

## [Author Response · Author response to Decision Letter 0]

30 Jul 2020

Journal Requirements:

We have tried to adhere to the specifications for the figure files but as some metrics are not completely unambiguous would request some guidance or assistance in case there are discrepancies.

2. At this time, we request that you please report additional details in your Methods section regarding animal care, as per our editorial guidelines:

(1) Please provide details of animal welfare (e.g., shelter, food, water, environmental enrichment)

A dedicated room to safely house immune compromised mice was used. Positive air flow and filtration was maintained though a BioBubble system. Mice were socially housed in groups of five within a micro isolation cage rack. Food and water were available ad lib. Each cage contained environmental enrichments, including nestlets and igloos.

(2) Please include the method of euthanasia 

Euthanasia was performed via exsanguination under anesthesia at the time of tumor extraction.

(3) Please describe the care received by the animals, including the frequency of monitoring and the criteria used to assess animal health and well-being during the course of the tumor study.

Tumor measurements and animal weights were collected on at least a weekly basis following the development of visible tumors. Animals were monitored daily with additional measurements taken if needed. Animals were euthanized if grafted tumor tissue shows signs of necrosis. Additional health signs were monitored with the following termination points: “rapid” weight loss of >10%, specific clinical signs of severe organ involvement (e.g. circling, head-tilt, respiratory rates, paralysis). Also, Body Condition Score (BCS) was monitored, with a score of 2 or below serving as a humane endpoint. 

The following clinical signs are indications of morbidity:

1. Persistent anorexia or dehydration

2. Consistent or rapid weight loss of 20% maintained for 72 hours.

3. Unable to maintain an upright posture or to move.

4. Muscle atrophy or emaciation.

5. Lethargy or failure to respond to gentle stimuli.

6. Hyperthermia.

7. Unconsciousness or coma.

8. Bloodstained or mucropurulent discharge from any orifice.

9. Labored respiration – particularly if accompanied by nasal discharge or cyanosis.

10. Anemia.

11. Significant abdominal distension.

12. Incontinence or prolonged diarrhea.

13. Dragging a limb or having trouble ambulating.

Animals were observed daily for signs or pain/distress. Animals that exhibited any of these behaviors were humanely euthanized.Thank you for your attention to these requests.

Changes were made to the methods section to include the details requested. We opted to omit the list of clinical signs of indications of morbidity we use (below) for conciseness but would be happy to include this as well if requested:

The following clinical signs are indications of morbidity:

1. Persistent anorexia or dehydration

2. Consistent or rapid weight loss of 20% maintained for 72 hours.

3. Unable to maintain an upright posture or to move.

4. Muscle atrophy or emaciation.

5. Lethargy or failure to respond to gentle stimuli.

6. Hyperthermia.

7. Unconsciousness or coma.

8. Bloodstained or mucropurulent discharge from any orifice.

9. Labored respiration – particularly if accompanied by nasal discharge or cyanosis.

10. Anemia.

11. Significant abdominal distension.

12. Incontinence or prolonged diarrhea.

13. Dragging a limb or having trouble ambulating.

3. Thank you for including your ethics statement: 'All animal studies were performed under an Institutional Animal Care and Use Committee–approved protocol.' 

(a) Please amend your current ethics statement to include the full name of the ethics committee/institutional review board(s) that approved your specific study. 

(b) Once you have amended this/these statement(s) in the Methods section of the manuscript, please add the same text to the “Ethics Statement” field of the submission form (via “Edit Submission”).

We believe this does not apply to our study since we used PDX models commercially available from Jackson Labs and an IRB was not required. Please advise if this is in conflict with your interpretation of this requirement. 

4. Please note that PLOS does not permit references to “data not shown.” Authors should provide the relevant data within the manuscript, the Supporting Information files, or in a public repository. If the data are not a core part of the research study being presented, we ask that authors remove any references to these data. Done

5. Please provide the product number and any lot numbers of the primary antibodies purchased from chemical companies for your study. 

A table has been added in the supplemental material listing this information.

6. To comply with PLOS ONE submission guidelines, in your Methods section, please provide additional information regarding your statistical analyses. For more information on PLOS ONE's expectations for statistical reporting, please see https://journals.plos.org/plosone/s/submission-guidelines.#loc-statistical-reporting. 

The Methods section has been augmented with the respective information

7. We note that you have indicated that data from this study are available upon request. PLOS only allows data to be available upon request if there are legal or ethical restrictions on sharing data publicly. For more information on unacceptable data access restrictions, please see http://journals.plos.org/plosone/s/data-availability#loc-unacceptable-data-access-restrictions. 

The respective data will be made public by uploading the entire RNASeq data set to an appropriate public server (see below).

N/A

In process; we will submit confirmation as soon as available.

8. Thank you for stating the following in the Competing Interests section:

WP, ES, FT, WSD and RB are employees of BD and FH and MF were BD employees

at the time the work was carried out.

We note that one or more of the authors are employed by a commercial company: BD Technologies and Innovation,

(I have corrected the exact wording)

(I have added an amended Funding Statement with the Cover Letter using other company papers as guidance for the wording)

 (I have completed using other company papers as guidance)

 (I have added an amended Competing Interest Statement with the Cover Letter using other company papers as guidance for the wording)

Response to Reviewers

5. Review Comments to the Author

Reviewer #1: In “High accuracy gene expression profiling of sorted cell subpopulations from breast cancer PDX model tissue”, Porter et al study the influence of fluorescence activated cell sorting (FACS) on transcriptional profiling and investigate the presence of multiple different subpopulations within the same breast cancer model tissue.” The manuscript seeks to understand whether there are consistent expression features of cell subpopulations in tumors, by using a xenograft model system. This is an important question for the field. While the techniques used are not at the forefront (scRNA-seq would be preferable), the concept of the work is sound. I have a few major concerns and some minor critiques.

Major

Line 180: The authors should provide a formal gene set analysis of the genes that were DE in both experiments, as well as for the DE genes unique to individual experiments. The analysis provided in lines 192-203 is relevant to this but is too inexact to be evaluated.

A gene set analysis in Enrichr was performed and supplied in the supplemental material along with a description in the results section.

The authors have a sizable number of RNA-seq replicates (line 132 suggests there are 32 RNA-seq datasets, though it is difficult to tell from the text) both across and within xenografts. However, there is little discussion of the robustness of the DE analysis across and within mice. This high number of replicates is the major strength of the work, so comparisons of replicates should be explicit.

Clarifying language was added in line 132 (caption Fig.1) and 137. 

The authors should describe how many cells were in each of the sorted populations. The proportions of cell populations that appear within xenograft ecosystems has been studied in pubs such as [Cell 167, 260–274.e22 (2016); Nat Comms 5, 5871 (2014); Sci Rep 8, 17937 (2018)] and may be relevant for tumor growth. Comments on this topic would strengthen the work. 

We appreciate this helpful suggestion. We provided the ratio of the respective supopulations in the supplemental data (S2) and added a paragraph about various forms of ITH (genotypic and phenotypic) and reports of outcomes of mixing cells with defined genotypic and or phenotypic characteristics. Since the primary aim of our work is in providing the most efficient path towards future functional studies of subpopulation interactions we refrain from drawing conclusions on the dynamic involved in our work for which we believe more than one time point would be required.

On a related point, are there published scRNA TNBC datasets available that the authors can compare to? Comparisons to such data would clarify the robustness of the observed effects.

We surveyed the literature specifically to address this point. There is no single repository with standard search protocols without which an exhaustive search would exceed our resources. We were however able to identify multiple examples of significant overlap between DEGs detected in scRNASeq data from TNBC and our gene lists. While we cannot exclude the possibility that additional relevant profiles exist we added three representative examples as supplemental figures (S9-S11) and edited the discussion accordingly.

Additionally we would like to point out that the gene set identified by Hassan et al. 2017 (see Fig. S6) in our original manuscript which correlates strongly with our data was identified in TNBC, albeit based on regulation within whole populations of cells.

Minor

Line 146: Please clarify that the references indicate the genes’ IEG status. This is not apparent from the table 1 caption.

The header of the table was changed to state “Reference identifying gene as IEG” as clarification.

Line 182: “120 of the 121 genes in common were qualitatively correlated (i.e. identical direction of change) and on a quantitative level the correlation coefficient was R= - 0.85” : The first and second parts of the sentence are stated inconsistently. If the direction of change is identical then the correlation coefficient should be positive. Rephrase.

We appreciate pointing out this error and have changed the wording to clearly state that it was an inverse correlation.

Some aspects of the article are out of date. E.g. Ref 12 lists a publication from 2017 as “in press.”

We have corrected the identified publication. Additionally we have replaced previous reference #9 with a peer reviewed publication by the same authors on the topic (previous reference was a pre-publication in Biorxiv).

Reviewer #2: Porter et al. have submitted a will written report investigating intratumoral heterogeneity of the transcriptome within solid human tumors by isolation of distinct cancer cell populations by FACS. Across tumor cases and replicates the authors identified immediate early gene changes, primarily due to tumor dissociation, that agreed to a high degree with previous publications. Cell populations were sorted based on high/low expression of the cell surface markers CD49f, CD133, and CD184, which were then analyzed by RNAseq. Differentially expressed genes were identified between populations indicating that each population may have distinct roles within the tumors. Overall, the experiments were designed well with proper controls, and this research will likely pave the way for more functional assays in the future (e.g. therapeutic response). I have no major concerns with the manuscript and have only a few minor comments/suggestions.

- In the Introduction I would suggest adding a brief description of the biological roles of CD49f, CD133, and CD184 and/or the types of cancer cell populations they have previously been used to isolate and define.

A passage has been added in the introduction including additional references to address this.

- For whole fresh tumor and the “bulk” dissociated cell population, what was the degree to mouse stromal (fibroblast, endothelial, and immune cell) cell infiltration? This can often be 10-50% in solid tumors. Did this have any effect on the RNAseq analysis for these samples?

We added the information in the results. Both models investigated by us so far contained 15% mouse cells which were excluded from the RNASeq analysis (in silico from the bulkk analysis, physically for “human only” and marker sorted cell populations) and to date we have not investigated differing models which would allow us to draw any conclusions in that regard.

---

## [Decision Letter · Decision Letter 1]

20 Aug 2020

High accuracy gene expression profiling of sorted cell subpopulations from breast cancer PDX model tissue

PONE-D-20-11376R1

Dear Dr. Blaesius,

We’re pleased to inform you that your manuscript has been judged scientifically suitable for publication and will be formally accepted for publication once it meets all outstanding technical requirements.

Kind regards,

Randall J. Kimple

Academic Editor

PLOS ONE

Additional Editor Comments (optional):

Reviewers' comments:

Reviewer's Responses to Questions

**Comments to the Author**

1. If the authors have adequately addressed your comments raised in a previous round of review and you feel that this manuscript is now acceptable for publication, you may indicate that here to bypass the “Comments to the Author” section, enter your conflict of interest statement in the “Confidential to Editor” section, and submit your "Accept" recommendation.

Reviewer #1: All comments have been addressed

Reviewer #2: All comments have been addressed

2. Is the manuscript technically sound, and do the data support the conclusions?

Reviewer #1: Yes

Reviewer #2: Yes

3. Has the statistical analysis been performed appropriately and rigorously? 

Reviewer #1: Yes

Reviewer #2: Yes

4. Have the authors made all data underlying the findings in their manuscript fully available?

Reviewer #1: Yes

Reviewer #2: Yes

5. Is the manuscript presented in an intelligible fashion and written in standard English?

Reviewer #1: Yes

Reviewer #2: Yes

6. Review Comments to the Author

Reviewer #1: I am satisfied with the revisions. All points have been addressed and I would recommend acceptance of the article.

Reviewer #2: All of my comments and concerns were addressed by the authors. However, I would suggest having the name of the repository of the uploaded NGS data in the final draft of the manuscript. Thank you.

7. PLOS authors have the option to publish the peer review history of their article (what does this mean?). If published, this will include your full peer review and any attached files.

Reviewer #1: No

Reviewer #2: No

---

## [Editor Report · Acceptance letter]

26 Aug 2020

PONE-D-20-11376R1 

High accuracy gene expression profiling of sorted cell subpopulations from breast cancer PDX model tissue 

Dear Dr. Blaesius:

I'm pleased to inform you that your manuscript has been deemed suitable for publication in PLOS ONE. Congratulations! Your manuscript is now with our production department. 

Kind regards, 

on behalf of

Dr. Randall J. Kimple 

Academic Editor

PLOS ONE